# Synthesis of CuO/GO-DE Catalyst and Its Catalytic Properties and Mechanism on Ciprofloxacin Degradation

**DOI:** 10.3390/nano12234305

**Published:** 2022-12-04

**Authors:** Ting Zhang, Jingjing Zhang, Yinghao Yu, Jinxu Li, Zhifang Zhou, Chunlei Li

**Affiliations:** School of Petrochemical Engineering, Lanzhou University of Technology, Lanzhou 730050, China

**Keywords:** diatomaceous earth, graphene oxide, copper oxide, ciprofloxacin, catalytic mechanism

## Abstract

A new catalyst, copper oxide/graphene oxide–diatomaceous earth (CuO/GO-DE), was prepared by the ultrasonic impregnation method. The optimal conditions for catalyst preparation were explored, and its structure and morphology were characterized by BET, XRD, SEM, TEM, FTIR, Raman and XPS. By taking ciprofloxacin as the target pollutant, the performance and reusability of CuO/GO-DE to degrade antibiotic wastewater was evaluated, and the optimal operating conditions were obtained. The main oxidizing substances in the catalytic system under different pH conditions were analyzed, as well as the synergistic catalytic oxidation mechanism. The intermediate products of ciprofloxacin degradation were identified by LC-MS, and the possible degradation process of ciprofloxacin was proposed.

## 1. Introduction

Ciprofloxacin (CIP), belonging to fluoroquinolones, is a common antibiotic [1,2], which is usually used to treat human or animal diseases, such as inhibiting the development of malignant cells or preventing humans or animals from being infected by bacteria [3,4,5,6]. However, due to the abuse of antibiotics [7] and the incomplete metabolism of antibiotics [8], CIP with a high concentration of about 10 μg/L has been detected in groundwater systems [9]. In effluents of hospitals and drug production facilities, the concentration of ciprofloxacin wastewater is as high as 150 μg/L, even more than 10 mg/L [10]. Antibiotic wastewater pollution has become very serious [11], which may bring significant harm to human health and the ecosystem [12], so it is urgent to explore efficient methods to treat ciprofloxacin wastewater [13]. At present, many treatment technologies have been developed, such as adsorption [14], biodegradation [15], advanced oxidation [16], etc. Heterogeneous catalysis (one of the advanced oxidation processes), as an efficient and environmentally friendly technology, has attracted much attention. Heterogeneous catalysts can catalyze H_2_O_2_ or H_2_O and generate radicals with strong oxidation ability, such as ·OH, ·O_2_^−^, and oxidize and decompose refractory organics to simple organics, even carbon oxide [5,9,11,16].

Diatomaceous earth (DE) is a biogenic siliceous sedimentary rock with active silica, light weight, and porous characteristics. The pores are regularly distributed, and the pore size ranges from more than ten to a hundred nanometers. As an important non-metallic mineral [17,18], diatomite has become a good adsorbent or catalyst carrier for functional materials [19,20,21] due to its large specific surface area, strong adsorption capacity, low price and rich reserves. Shen [22] et al. applied diatomite modified with NaOH and MnCl_2_ to adsorb heavy metal ion Cd(II) in wastewater, and the adsorption ratio of Cd(II) reached 98.69%. Xiong [23] et al. prepared Ag_3_PO_4_/Fe_3_O_4_/DE composites by one-step hydrothermal method and studied their photocatalytic properties, and the results showed that when the content of Fe_3_O_4_ was 8%, the degradation ratio of pollutants reached 98%. Sun [24] et al. prepared porous MnFe_2_O_4_/diatomite (MFD) material by solvothermal method and explored the degradation effects of tetracycline hydrochloride (TC-HCl) by MFD, and the results showed that the material had a good catalytic effect on TC degradation at pH ranging from 3 to 11. Dang [25] et al. prepared a diatomite composite coated with amorphous manganese oxide (MnO_2_) in acidic conditions, which showed high efficiency in the degradation of methylene blue and methyl orange. Wang [26] et al. mixed g-C_3_N_4_ with diatomite to prepare CN/DE-10 composite and explored its degradation ability on Rhodamine B (RhB), which can be effectively degraded within 50 min. Daniel [27] et al. prepared spherical diatomite Fenton catalyst, and the degradation ratio of clindamycin reached 89.7% at pH 3.

Graphene oxide (GO), first discovered in 1859, is an ideal single-layer amorphous two-dimensional material [28,29]. As a derivative of graphene, it retains the carbon skeleton and the same properties of graphene [30], while it is easier to prepare than graphene [31]. Therefore, GO is widely used in many fields. It has been proven to have practical and efficient applications in the fields of environmental catalysis [32], functional materials [33] and conductivity [34,35]. Hassan [36] et al. prepared RGO/SZ material by loading silica and zirconia on reduced graphene oxide and explored its degradation ability of bisphenol A (BPA). The results showed that under optimal conditions, the degradation ratio for BPA reached 88%. Liu [37] et al. prepared GO-SH/DE composite adsorption material through the combination of diatomite (DE) and thiolated graphene oxide (GO-SH), and the adsorption capacity of the material for patulin (PAT) in apple juice was explored. Under the best conditions, the adsorption capacity for PAT reached 90%. Metal oxides (Fe_2_O_3_, CeO_2_, CuO, etc.) can be employed as active components coated on catalyst supports because they have similar catalytic effects to Fe^2+^ or Cu^2+^ [38,39]; moreover, they are not easy to lose and can be used in a wide pH range. In this study, diatomaceous earth (DE) was modified by graphene oxide (GO) to improve the specific surface area and enhance surface capacity, and then the modified diatomite carrier was loaded with different active metal salts separately to screen the best active component (metal oxide) for CIP degradation. An optimized CuO/GO-DE catalyst was prepared, and the degradation performance and mechanism of ciprofloxacin (CIP) were studied.

## 2. Experimental Section

### 2.1. Materials

Diatomite was purchased from Aladdin Company (Los Angeles, CA, USA), with a silica content of 90%, alumina content of 3%, and magnesium oxide and calcium oxide content of 0.5%; Nano graphite powder and ciprofloxacin were also purchased from Aladdin company. Iron nitrate, copper nitrate, nickel nitrate, cerium nitrate, silver nitrate and sodium nitrate were purchased from Tianjin Kaixin Chemical Co., Ltd. (Tianjin, China) Potassium permanganate, concentrated hydrochloric acid, concentrated sulfuric acid and hydrogen peroxide were purchased from Shanghai Sinopharm Chemical Reagent Co., Ltd. (Shanghai, China). All chemical reagents were analytically pure and had not been further purified before use. In this study, the experimental water is distilled water, which is self-made by our laboratory.

### 2.2. Synthesis of Metal-Loaded GO-DE Composites

Graphene oxide (GO) was prepared by the Hummers method [38,40]. A total of 1 g graphite powder and 1 g sodium nitrate were added into 60 mL concentrated sulfuric acid and stirred vigorously in a low-temperature water bath, then 6 g potassium permanganate was slowly added and stirred for 3 h. The reaction mixture was put into a water bath with a temperature of 35 °C and continuously stirred for 3 h, then 140 mL of deionized water was added and stirred at room temperature for 12 h. A total of 200 mL of deionized water was added into the mixed solution, then 20 mL of hydrogen peroxide was slowly added, stirred for 1 h and stood for 5 h. After the product was settled, it was washed with dilute hydrochloric acid, then washed with deionized water to neutral, and ultrasonic for 2 h at 100 Hz to obtain a viscous graphene oxide solution.

Some distilled water was added to the viscous GO solution, and GO dispersion liquid was obtained by 100 Hz ultrasound. Different dosages of purified DE were added to the above GO dispersion liquid; the mixture was magnetically stirred and then centrifuged. After pouring out the supernatant, the pasty clay was fully stirred again and then dried in an oven. The sample was then ground and sieved by a 100-mesh sieve, and a GO-DE sample was obtained.

Metal-loaded GO-DE composite was prepared by ultrasonic dipping method. Five different metal salts (copper nitrate, iron nitrate, nickel nitrate, cerium nitrate and silver nitrate) were used as predecessors. They were dissolved in distilled water, respectively, and the prepared GO-DE was put into a different solution and ultrasonic impregnated for 30 min with 0.1 M NaOH solution added. Then the sample was filtered, separated, washed with distilled water 5 times, and dried at 110 °C for 2 h to obtain five different metal-loaded GO-DE composites, which were used as heterogeneous catalysts in our study.

### 2.3. Characterization

The specific surface area, pore volume and pore diameter of the material were measured at −196 °C with ASAP 2010 specific surface area meter produced by American Micromitrics Company (Atlanta, GA, USA). The surface morphology of the samples was observed by JEOL JSM-6701F scanning electron microscope (SEM) with an accelerating voltage of 20 kV. The microstructures were investigated by JEOL JEM-1200EX (JEOL Ltd., Tokyo, Japan) transmission electron microscope (TEM). Before SEM and TEM tests, the samples need to be ground and ultrasonically dispersed in an ethanol solution. The content and mapping of the main elements of the materials were tested and analyzed by energy spectrometer (EDS) from Brooke Technology Co., Ltd., (Beijing, China) Nicolet AVTAR 360 FT-IR infrared spectrometer (Thermo Electron Co., Waltham, MA, USA) was used to test the chemical structure, including the changes in the compositional or functional group of samples, with the scanning range from 4500 cm^−1^ to 400 cm^−1^. The molecular structures of the samples were tested by a PERS-SR532 Raman spectrometer (Persetech Company, Xiamen, China). Crystal structures of the samples were identified by Panalytical X’Pert PRO X-ray diffractometer. The working voltage and current were 40 kV and 150 mA, the diffraction angle was 5° to 80°, and the scanning step was 0.02°/s. The PHI5702 X-ray photoelectron spectrometer (American Physical Electronics Company, Chanhassen, MN, USA) was used to quantitatively and qualitatively analyze the elements on the material surface and the chemical valence state and valence electron state of the elements. The X-ray emission sources are Mg, Al double anode target and Al monochromator target.

### 2.4. CIP Degradation

A total of 250 mg/L CIP stock solution was prepared for later use. Additionally, 20 mL CIP stock solution was taken in a conical flask and 80 mL distilled water was added to obtain 100 mL 50 mg/L CIP solution. The pH value of the solution was adjusted with sulfuric acid (1:9) or NaOH (0.1 mol/L), and a certain amount of catalyst and hydrogen peroxide were added into the conical flask. The conical flask was then placed in a thermostatic oscillator with a water bath at the speed of 120 rpm. The supernatant was taken at regular intervals to measure the absorbance by UV-1900 ultraviolet-visible spectrophotometer at λ of 277 nm, at which CIP has the maximum absorption peak. The degradation ratio of the CIP solution can be calculated by the following formula: η = (C_0_−C_t_)/C_0_ × 100%, in which C_0_ is the initial concentration of CIP solution, and C_t_ is the concentration of CIP solution with the reaction time t.

### 2.5. Measurement of ·OH Concentration and ·OH Scavenging Experiment

·OH reacts with salicylic acid to produce 2,3-dihydroxybenzoic acid [39], which has the maximum absorption peak at the wavelength of 510 nm. ·OH concentration can be measured by the following methods: 100 mL salicylic acid solution with a certain concentration was put into a 250 mL conical flask, and 1 g/L catalyst and 0.1 mL hydrogen peroxide were added into it. The conical flask was put into a water bath at 60 °C. With the reaction happening, the ·OH concentration of the solution can be measured at regular intervals by UV-1900 ultraviolet-visible spectrophotometer at 510 nm.

Tert-butyl alcohol (TBA) is a strong ·OH scavenger [41], which can effectively inhibit the occurrence of the Fenton reaction. The CIP solutions (100 mL, 50 mg/L) were prepared under pH 4, 7 and 10 separately, and the catalyst, hydrogen peroxide and excessive TBA were also added to the solution. The control group was set, the conical flask was put into a water bath at 60 °C for reaction, and the degradation ratios of CIP were tested, respectively, when the reactions were all balanced.

### 2.6. Capture of Cu(III) in Alkaline Condition

According to the literature reports, when periodate is added to the alkaline reaction system, it can form Cu(III)—periodate complex to stabilize the unstable Cu(III) in the reaction system [42,43]. Based on the experimental scheme of Li et al. [43], periodate was added to the alkaline catalytic reaction system containing CuO/GO-DE compound material, hydrogen peroxide and NaOH, and the control group was set. After reacting in a water bath thermostatic oscillator for 5 min, the water sample was taken out and filtered through a 0.22 µm organic filter membrane, and the absorbance of the filtered solution was measured by UV-1900 ultraviolet-visible spectrophotometer at 415 nm.

## 3. Result and Discussion

### 3.1. Optimization of Catalyst Preparation

Diatomite (DE) does not have a high specific surface area and good adsorption capacity as activated carbon, and it cannot be used as an absorbent directly. In order to take good advantage of DE, graphene oxide (GO) was introduced into diatomite to improve its adsorption property. The removal efficiencies of CIP by diatomite modified with different GO proportions were compared, and the experimental conditions were as follows: catalyst dosage of 2 g/L, pH value of 7, reaction temperature of 60 °C, CIP initial concentration of 50 mg/L, the results were shown in Figure 1a. It can be found that the removal ratios of CIP rise when GO proportions increase from 3% to 10%. Therefore, the more GO is introduced into DE, the higher the adsorption efficiency of the material. As we know, The preparation of GO is relatively complex, costly and difficult to produce in large quantities; considering the low cost and time saving, the proportions of GO around 7~10% integrated with DE was better for this composite support’s preparation.

The activities of different metal oxides loaded on GO-DE and their catalytic properties were also studied. CuO, Fe_2_O_3_, NiO_2_, Ag_2_O and CeO_2_, which are five types of metal oxides (metal nitrates were used as the precursors), were selected as active substances and coated on GO-DE support; the CIP degradation capacities of these catalytic composite materials can be seen in Figure 1b. According to Figure 1b, the GO-DE support loaded with CuO or Ag_2_O have the best degradation abilities of CIP, while Ag or Ag salt are all not cheap. From economic considerations, copper nitrate is the best one for GO-DE-based catalyst preparation. Not only the type of metal oxides but also the loading amount of it on the catalyst surface influences the catalytic efficiency of the catalyst. Different concentrations of precursor Cu(NO_3_)_2_ solutions were used to prepare the catalysts, which were applied for CIP degradation. It can be seen from Figure 1c that the different precursor concentration for the catalyst preparation has different catalytic performance on CIP degradation, and when using 5% Cu(NO_3_)_2_ precursor concentration to prepare the catalyst, the CIP degradation ratio is the highest.

### 3.2. Characterization of CuO/GO-DE

In order to observe the surface morphology of the catalyst and verify the successful loading of copper oxide on GO-DE support, BET test, SEM, XRD, FT-IR and XPS were used to analyze it.

Table 1 shows the comparison of BET-specific surface area measurement results of DE, GO-DE and CuO/GO-DE. It can be seen from Table 1 that DE does not have a high specific surface area, while adding GO into DE can greatly improve the specific surface area of DE, nearly double that of DE. The specific surface area of CuO/GO-DE is reduced a little due to the metal loading. Total pore volume and average pore diameter have changed but not too much when introducing GO and CuO into DE.

Figure 2a–c shows the SEM images of DE, GO-DE and CuO/GO-DE, respectively. From Figure 2a, holes with diameters from 0.05 to 0.6 μm on the diatomite surface can be seen clearly. Figure 2b shows that the surface of DE is evenly covered by folded thin-film GO with the holes preserved, and the diameter of the holes is even higher than before (0.05–0.75 μm), which can effectively improve the specific surface area of diatomite. After being coated by CuO, the holes become smaller than before (Figure 2c). Figure 2d shows the TEM diagram of CuO/GO-DE; it can be clearly seen that columnar CuO is embedded in GO. In addition, Figure 2e shows EDS mapping images of CuO/GO-DE composites. It can be seen that the composites contain Si, C, O and Cu elements. Si comes from DE, C, O from GO and DE, and Cu from loaded metal copper salt. It showed that the Cu element was loaded on the composite material GO-DE successfully and uniformly, which plays an important part in the catalytic performance of the prepared catalyst.

The FTIR spectra of GO, DE, GO-DE and CuO/GO-DE are shown in Figure 3a. In the spectra of GO and DE, the broad band absorption peaks at 3440 cm^−1^ are related to the surface hydroxyl stretching vibration of GO and DE, respectively [41,43]. The peak at 1623 cm^−1^ is attributed to the stretching vibration of the C=C bond on GO, and the peak at 1087 cm^−1^ is the stretching vibration of the C-O-C bond on GO [44]. The peak at 1094 cm^−1^ is attributed to the symmetric stretching vibration of the Si-O-Si bond of DE; The peak at 794 cm^−1^ is attributed to the vibration of the Si-OH bond of DE [23,45]. GO-DE and CuO/GO-DE have a C=C bond and C-O-O bond as GO has, which proves the existence of GO in composite materials. In CuO/GO-DE spectrum, there is a weak absorption peak at 620 cm^−1^, related to the vibration of the Cu(II)-O bond [46].

Raman spectroscopy can be used to characterize the defects of GO. In general, the Raman spectrum of GO has two characteristic peaks: G and D peaks. The G peak, caused by the first-order scattering of E_2g_ lattice vibration by sp^2^ carbon atom hybrid vibration, is around 1580 cm^−1^; the D peak, caused by sp^3^ hybrid carbon atoms, is about 1350 cm^−1^ [47,48]. Figure 3b shows the Raman characterization of DE, GO-DE and CuO/GO-DE, from which it can be clearly seen that there are two characteristic peaks (the D peak at 1354 cm^−1^ and the G peak at 1588 cm^−1^) of GO on the GO-DE support and CuO/GO-DE composite, GO is successfully integrated with DE.

The XRD patterns of GO, DE, GO-DE and CuO/GO-DE are shown in Figure 3c. The diffraction peaks at 2θ = 21.80°, 26.04°, 29.01°, 32.56°, 46.78° and 57.23° correspond to the crystal structure of DE [49,50]. It can be seen that the preparation of CuO/GO-DE did not change the structure of DE. Jade6 XRD software was used to analyze weak diffraction peaks at 2θ = 39.28°, 41.34°, 58.62° and 68.36° of CuO/GO-DE sample, which corresponds to 45-0937 card of the standard sample, indicating those peaks are the characteristic diffraction peaks of CuO [40,51]. It was confirmed that CuO was successfully loaded on the GO-DE carrier. GO has a strong characteristic peak at 2θ = 11.42° [52], but the peaks are very weak in the spectra of GO-DE and CuO/GO-DE, which may be due to the small proportion of GO in the composite.

In order to make sure the chemical state of various elements in the sample, the CuO/GO-DE was tested by XPS. The total spectrum and the characteristic spectrums of each element are shown in Figure 3d–h. In the total spectrum (Figure 3d), it can be seen that the peaks near the binding energies of 103 eV and 150 eV correspond to the characteristic peaks of Si 2 p and Si 2 s: the peak with binding energy around 285 eV belongs to the characteristic peak of C 1 s; the peak of binding energy near 532 eV is the characteristic peak of O 1 s; the peaks of binding energy near 933 eV and 953 eV are the characteristic peaks of Cu 2 p3/2 and Cu 2 p1/2, respectively, indicating the existence of divalent copper (CuO) in the catalyst. Figure 3e shows the characteristic peaks at the binding energies of 284.8 eV, 286.9 eV and 288.9 eV corresponding to C-C, C-O/C=O and O-C=O [47,51], respectively. Figure 3f shows the spectrum of O 1 s with the peaks at 531.65 eV and 532.7 eV representing C=O of GO and the peaks at 533.5 eV representing C-O [38,46]. In Figure 3g, the peak at the binding energy of 103.1 eV corresponds to the characteristic peak of Si 2 p, indicating the main components of DE. The spectrum of Cu 2 p in the sample (Figure 3h) has the characteristic peaks of Cu 2 p3/2 and Cu 2 p1/2 at the binding energies of 933 eV and 953 eV, respectively [52], indicating that CuO exists on the surface of the catalyst and the copper is loaded on the material in the form of CuO.

### 3.3. Degradation of CIP

In order to determine the optimum operating conditions during CIP degradation, a series of single-factor experiments were carried out at different pH values, initial CIP concentration, catalyst dosage, H_2_O_2_ concentration and reaction temperature.

pH value is an important factor affecting the catalytic performance of CuO/GO-DE. Figure 4a shows the degradation capacity of the catalyst under different pH conditions. It can be seen that the catalytic system achieved high degradation efficiencies at pH ranging from 4 to 10, and at pH of 7, it can reach the highest CIP degradation ratio (99.9% after 240 min of reaction). Under acidic conditions, ·OH generated from catalyzing H_2_O_2_ by CuO/GO-DE is the main oxidant to oxidize the CIP under neutral and alkaline conditions, although the amount of ·OH reduced due to H_2_O_2_ decomposing to H_2_O and O_2_, Cu(I) will be oxidized by H_2_O_2_ and O_2_ to high valence Cu(III) with strong oxidizability [53,54,55,56], which plays an important role as well as ·OH does in the degradation of CIP.

The results fully showed that the catalyst could degrade ciprofloxacin in a wide pH range with H_2_O_2_. The traditional Fenton oxidation method is applicable to a very narrow pH range of 2.5~3.5. However, the pH of most organic wastewater is near neutral. The CuO/GO-DE catalyst has great advantages over the traditional Fenton catalyst.

When the concentration of pollutants is certain, the relationship between catalyst dosage and H_2_O_2_ dosage is mutual restriction. It can be seen from Figure 4b of the experimental results that it is not true; the more catalyst or H_2_O_2_, the better the catalytic effect. There should be a certain proportional relationship between the two factors. In other words, the active site (coated CuO) of the catalyst and H_2_O_2_ should be in a certain proportion in order to achieve the optimal catalytic effect. The experimental and calculation results showed that the catalytic effect is the best when the mass ratio of the catalyst to H_2_O_2_ is 0.133.

As can be seen from Figure 4c, with the increase in reaction temperature, the degradation ratio of ciprofloxacin gradually increased, and the degradation rate and efficiency reached the maximum at 50 °C. When the reaction temperature was increased to 60 °C, the degradation efficiency did not rise, but the degradation rate decreased. The reason is that with the increase in the temperature, the reaction activation energy increases, which accelerates the catalytic reaction speed and increases the amount of ·OH and Cu(III), thus improving the efficiency of the catalytic system for the degradation of CIP. The too-high temperature accelerates the decomposition of H_2_O_2_, so 50 °C is the best temperature for the catalytic reaction.

The recyclability of the CuO/GO-DE catalyst was studied through continuous reusability experiments. The catalyst was reused for the next run after washing it with distilled water without further treatment. The experimental results are shown in Figure 4d. After five cycles, the degradation ratio of ciprofloxacin decreased from 99.9% to 80%, which showed a certain stability of the catalyst. The reduction in degradation ratio may be because of the degradation product residue on the catalyst surface, which reduces the surface catalytic active sites.

### 3.4. Degradation Mechanism of Ciprofloxacin

In order to analyze the CIP degradation process and mechanism of the catalytic system, comparative experiments of CIP treatment in different systems, ·OH concentration, Cu(III) measurements of different systems and ·OH extinction experiment were carried out.

The comparative experiments were performed when (1) only H_2_O_2_ existed, (2) only CuO/GO-DE composite existed or (3) both CuO/GO-DE and H_2_O_2_ existed. The results are shown in Figure 5a. When there was only H_2_O_2_ in the system, just part of CIP was oxidized and degraded, and the removal ratio was low. When there was only CuO/GO-DE in the system, it can be seen that the removal ratio of CIP can reach about 85%. This is because, in the case of the absence of H_2_O_2_, CuO/GO-DE may catalyze O_2_ to ·O_2_^−^, which also has an effect on the oxidative degradation of CIP. When there were both CuO/GO-DE and H_2_O_2_ in the system, the degradation rate of CIP jumped to 99.9%, indicating that CuO/GO-DE/H_2_O_2_ system has stronger oxidation than other systems.

In order to explore the main oxidizing substances in the reaction system, the concentration of ·OH and the content of Cu(III) in the system were determined. The results are shown in Figure 5b. It can be seen that under the conditions of catalyst dosage 1 g/L, H_2_O_2_ concentration 3.916 mmol/L, reaction temperature 50 °C, pH value 7 and reaction time 240 min, there were a lot of Cu(III) and ·OH produced in CuO/GO-DE/H_2_O_2_ system, while there were almost no Cu(III) and ·OH produced in CuO/GO-DE and H_2_O_2_ systems separately.

In order to further determine the contributions of Cu(III) and ·OH in the degradation of CIP in this heterogeneous system, the ·OH scavenging experiments were carried out, and the results are shown in Figure 5c. It can be seen that when the pH value was 4, the addition of tert-butyl alcohol significantly inhibited the produced ·OH as well as the CIP degradation ratio, which dropped from 86.5% to 17.6%. Under acidic conditions, copper existed in the form of Cu(I) or Cu(II) and catalyzed H_2_O_2_ to generate a large amount of ·OH. At this time, the reaction system mainly relied on ·OH to oxidize and decompose the target pollutants. At a pH of 7, the degradation ratio of CIP decreased from 99.9% to 68.3% with tert-butyl alcohol. The inhibition of tert-butyl alcohol on ·OH and CIP degradation ratio was significantly weakened, indicating that in addition to ·OH, there were other strong oxidizing substances, such as Cu(III), that play the role of oxidative decomposition to target pollutants at this time. At pH of 10, the degradation ratio of CIP decreased from 93.5% to 85.7% with tert-butyl alcohol, and the inhibition of tertbutyl alcohol on the degradation ratio was further weakened, indicating that at this time, there was little ·OH obtained from the decomposition of H_2_O_2_ in the solution, and Cu(III) was the main oxidant for oxidative decomposition of target pollutant. From the experimental results, it can be seen that under medium and alkaline conditions, Cu(II) is oxidized to higher valence Cu(III), and as the pH value rises, the number of Cu(III) is significantly increased, reflecting that Cu(III) is strongly dependent on pH value [41,57,58,59]. In summary, we can infer that CuO/GO-DE materials have different degradation modes of CIP at different pH values: when the solution is acidic, the degradation of CIP mainly depends on the oxidation of ·OH; when in neutral, it depends on the joint action of Cu(III) and ·OH; when in alkaline, it mainly depends on the oxidation of Cu (III). This explains why the catalyst has excellent degradation ability for CIP in a wide pH range due to the synergistic effect of ·OH and Cu(III). CIP removal ratios at pH values of from 2 to 12 were also tested in CuO/GO-DE/H_2_O_2_ system and CuO/GO-DE/H_2_O_2_ + tert-butyl alcohol system, and the contribution curves of Cu(III) + ·OH and Cu(III) to CIP removal ratios varied with pH was obtained, and the contribution curve of ·OH can also be deduced, as can be seen in Figure 5d.

According to the morphology analysis and mechanical analysis, the preparation process of Cu/GO-DE and degradation mechanisms of CIP by Cu/GO-DE/H_2_O_2_ system in different pH ranges were shown in Figure 6a,b separately.

In addition, the intermediate products of CIP degradation were described by LC-MS analysis, and the possible degradation pathway of the CIP molecule was speculated, as shown in Figure 7. The intermediate P1 (*m*/*z* = 362) was caused by the attacking of active oxides in the reaction system to the piperazine ring on the CIP molecule; P2 (*m*/*z* = 308) is diethylene CIP, which was produced by losing two C = O bonds of P1, and P3 (*m*/*z* = 263) is aniline, which was formed due to the loss of C_2_H_5_N of P2 [38]. P3 may also be formed due to the piperazine ring directly falling off caused by the active oxide’s attack on the CIP molecule [60]; P3 then lost the cyclopropyl and amino group to obtain P4 (*m*/*z* = 156) [61], and the fluorine atom on P4 was replaced by -OH and the quinolone ring was cracked to form P5 (*m*/*z* = 154). In addition, the intermediate product P6 (*m*/*z* = 290) was formed by ring cracking due to the oxidation of the cyclopropyl group on the CIP molecule. The active oxides in the reaction system then continued to attack the piperazine ring and quinolone ring on the CIP molecule, resulting in its cracking and the formation of P8 (*m*/*z* = 152). P5 and P8 may be further transformed into low molecular organics and eventually mineralized into water and carbon dioxide.

## 4. Conclusions

In this study, a copper-loaded graphene–diatomaceous earth (CuO/GO-DE) catalyst was prepared. By changing the loading ratio of graphene oxide and copper, the optimal conditions for the preparation of the catalyst were explored. The catalyst was characterized by BET, SEM, TEM, Raman, FTIR, XRD and XPS, which showed that introducing GO into DE optimized the surface morphology and microscopic properties of DE and confirmed that copper oxide was successfully and evenly coated on GO-DE support. The effects of pH value, CIP initial concentration, reaction temperature, and catalyst and H_2_O_2_ dosages on CIP degradation by CuO/GO-DE catalytic system were studied. The results showed that CuO/GO-DE composites had excellent catalytic degradation activity in a wide range of pH. Under the best degradation conditions (pH of 7, CIP initial concentration of 50 mg/L, mass ratio of catalyst to H_2_O_2_ of 0.133, reaction temperature of 50 °C), the degradation ratio of CIP can reach 99%. The catalyst could be used repeatedly, and the degradation catalytic efficiency did not decrease significantly. Through the determination or quenching experiments of ·OH, Cu(III), etc., the degradation mechanisms of CuO/GO-DE catalyst for CIP at different pH values were proposed: under acidic conditions, the CIP degradation mainly depends on the oxidation of ·OH; under neutral conditions, it depends on the synergistic oxidation of Cu(III) and ·OH; under alkaline conditions, it mainly depends on the oxidation of Cu(III). The intermediate products of CIP degradation were identified by LC-MS. According to the main identified products, the possible degradation process of CIP in the catalytic system was proposed. This study provided a simple and effective method for the degradation of antibiotics in wastewater.

## Figures and Tables

**Figure 1 nanomaterials-12-04305-f001:**
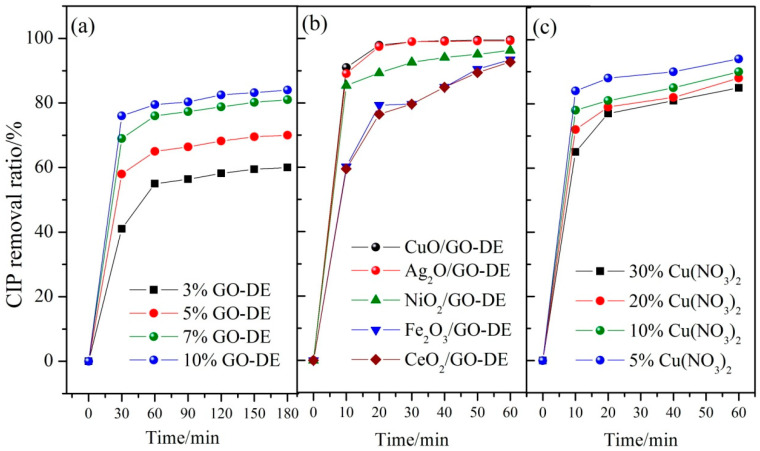
(**a**) Influence of proportions of GO integrated with DE on CIP removal ratio; (**b**) influence of metal type loaded on GO-DE on CIP removal ratio; (**c**) influence of precursor concentration on CIP removal ratio.

**Figure 2 nanomaterials-12-04305-f002:**
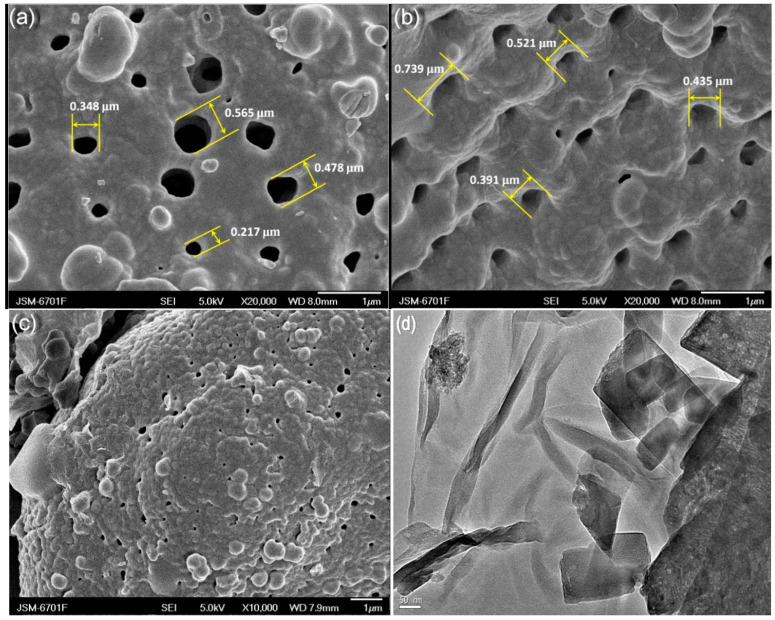
SEM micrographs of (**a**) DE, (**b**) GO-DE, and (**c**) CuO/GO-DE; (**d**) TEM micrograph of CuO/GO-DE; (**e**) EDS mapping images of CuO/GO-DE composites.

**Figure 3 nanomaterials-12-04305-f003:**
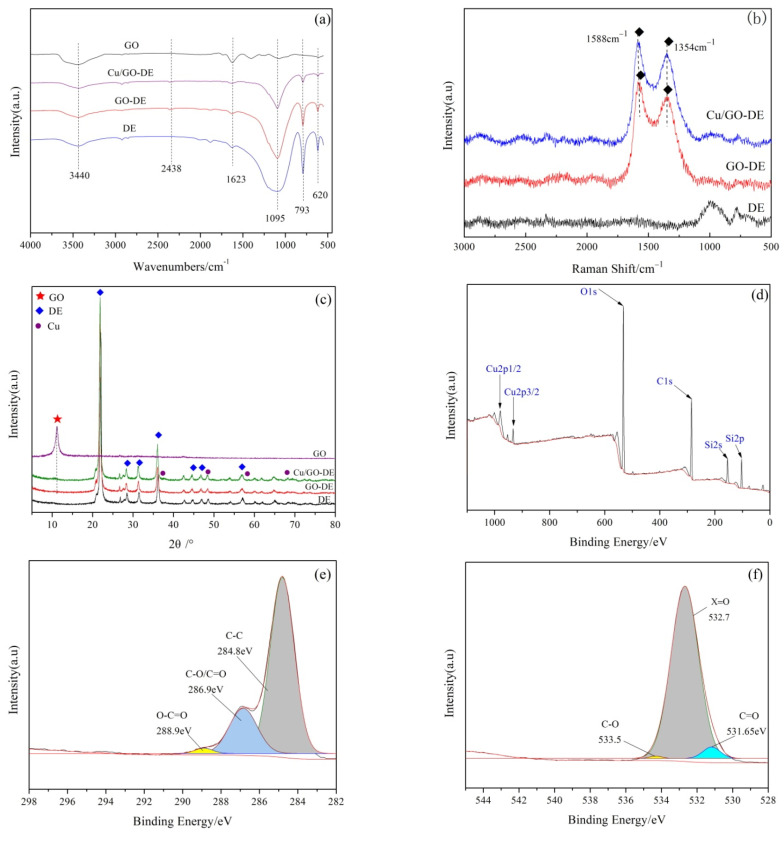
(**a**) FTIR spectra of DE, GO, GO-DE and CuO/GO-DE; (**b**) Raman spectra of DE, GO-DE and CuO/GO-DE; (**c**) XRD patterns of DE, GO, GO-DE and CuO/GO-DE; XPS spectra of CuO/GO-DE: (**d**) total spectrum, (**e**) C 1 s, (**f**) O 1 s, (**g**) Si 2 p and (**h**) Cu 2 p.

**Figure 4 nanomaterials-12-04305-f004:**
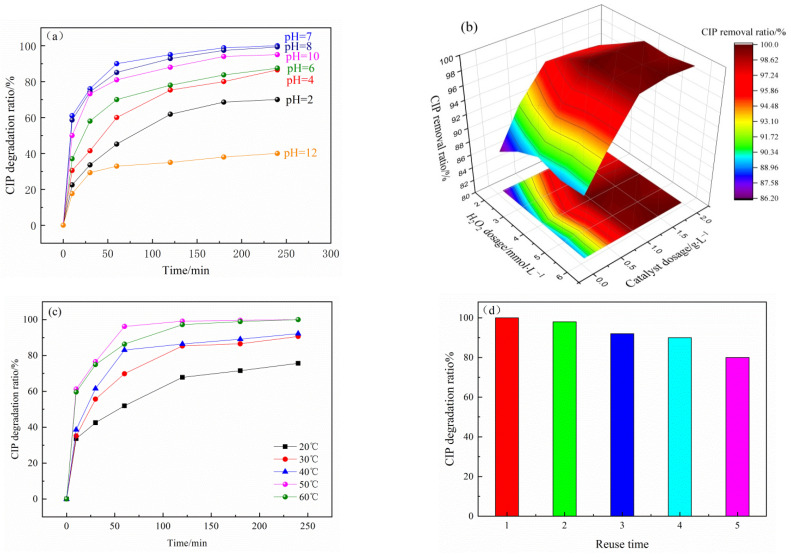
Effects of pH value (**a**), catalyst dosage and H_2_O_2_ concentration (**b**); temperature (**c**) on the degradation of CIP by CuO/GO-DE catalyst; the reusability of CuO/GO-DE catalyst (**d**).

**Figure 5 nanomaterials-12-04305-f005:**
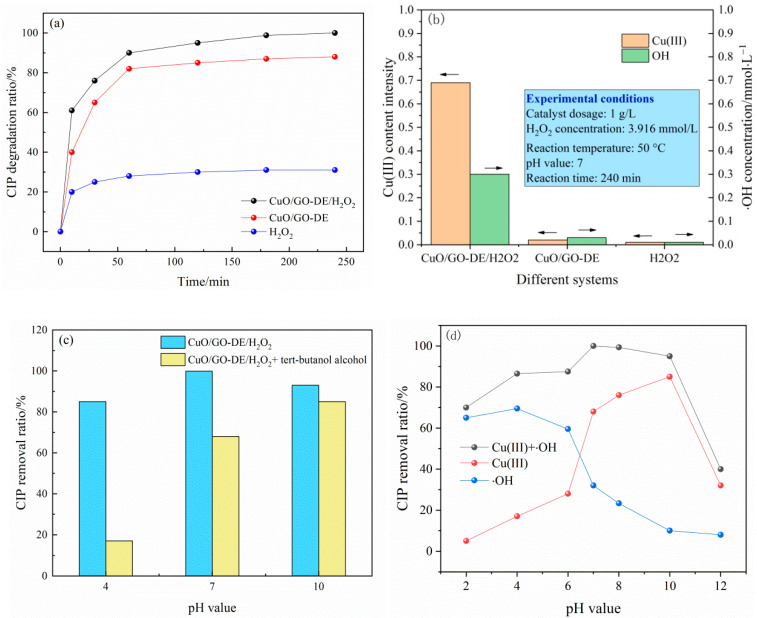
(**a**) Removal of CIP by different systems; (**b**) ·OH concentration and Cu(III) measurement in different systems; (**c**) ·OH scavenging experiments; (**d**) contribution of ·OH and Cu(III) on CIP removal ratios in different pH value.

**Figure 6 nanomaterials-12-04305-f006:**
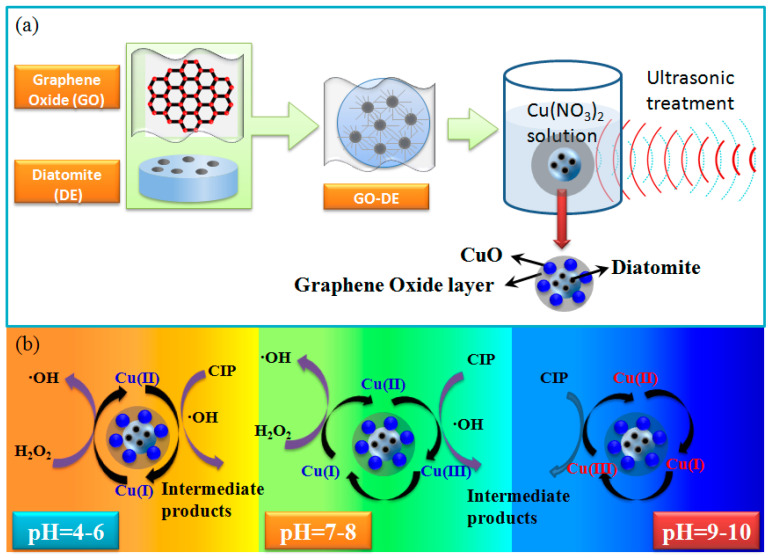
Preparation process of Cu/GO-DE (**a**) and degradation mechanism of CIP by Cu/GO-DE/H_2_O_2_ in different pH ranges (**b**).

**Figure 7 nanomaterials-12-04305-f007:**
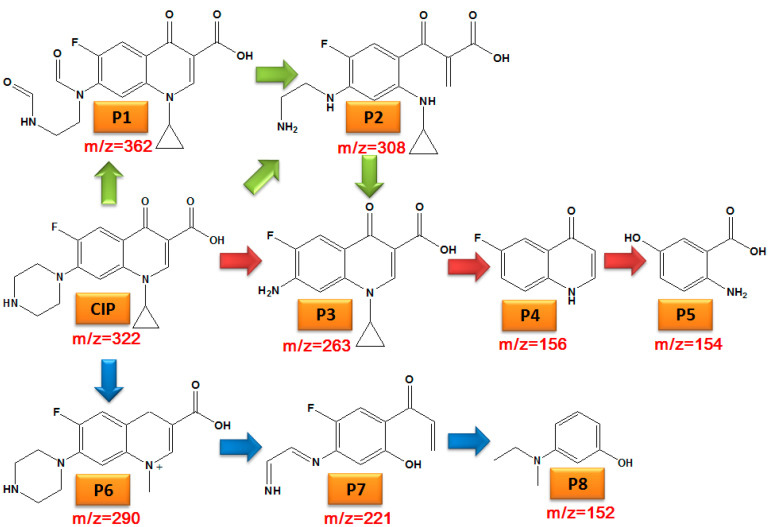
Possible degradation pathway of CIP.

**Table 1 nanomaterials-12-04305-t001:** BET measurement results of DE, GO-DE and Cu/GO-DE.

Samples	BET Specific Surface Area (m^2^/g)	Total Pore Volume (cm^3^/g)	Average Pore Diameter (μm)
DE	10.7239	0.021179	0.483013
GO-DE	18.7833	0.027155	0.553611
CuO/GO-DE	15.4166	0.025061	0.526992

## Data Availability

Data available on request from the authors.

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
