# Peer review of "Synthesis of CuO/GO-DE Catalyst and Its Catalytic Properties and Mechanism on Ciprofloxacin Degradation"

_nanomaterials, 2022, doi:10.3390/nano12234305_

Round 1
Reviewer 1 Report
The paper is about the deep study of a composite material of Diatomite-based catalyst for de degradation of ciprofloxacin. The paper is well written and scientifically interesting, and the material and the catalytic properties are well studied and explained, but some minor revisions and/or clarification are mandatory.
1) In the experimental section line 80-81-88 volumes or quantities used are not precisely reported, but more like "a certain amount." This way of reporting does not sound very scientific.
2)Figure 1, in the caption it is written that the data are relative to the CIP removal, but in the text figure 1 b and c are associated with the degradation of MB, please correct or clarify.
3) Figure 1 c) in the text is reported that the maximum degradation of MB is obtained with 10%Cu(NO3)2 but in the relative graph is reported the CIP degradation and the maximum are reached using the 5% loaded Cu(NO3)2. Please explain this point.
4)Figure 2 (b) in the text is reported that the folded thin film of GO is evenly covered on the surface of DE, I do not understand from what this statement can be deduced, seeming that images 1(a) and 1(b) are the same
Author Response
18-Nov-2022
Manuscript ID: nanomaterials-2033423
Title: Study on morphology regulation of a diatomite-based catalyst and its catalytic mechanism on degradation of CiprofloxacinNanomaterials
Responses to reviewer 1’s comments:
Thanks for your useful comments and suggestions of our manuscript. We have modified the manuscript accordingly, and the detailed corrections are listed below point by point:
1) In the experimental section line 80-81-88 volumes or quantities used are not precisely reported, but more like "a certain amount." This way of reporting does not sound very scientific.
---We have changed the statements like “a certain amount”.
2) Figure 1, in the caption it is written that the data are relative to the CIP removal, but in the text figure 1 b and c are associated with the degradation of MB, please correct or clarify.
---Sorry, we made a low-level mistake, and we have corrected them.
3) Figure 1 c) in the text is reported that the maximum degradation of MB is obtained with 10%Cu(NO3)2 but in the relative graph is reported the CIP degradation and the maximum are reached using the 5% loaded Cu(NO3)2. Please explain this point.
---We made the experiments of influences of Cu(NO3)2 contents on both MB and CIP degradations. When degrading MB, 10% Cu(NO3)2 is best, while degrading CIP, 5% Cu(NO3)2 is best. Sorry, we really made low-level mistake, we put on the right figure, but wrong explanation.
4)Figure 2 (b) in the text is reported that the folded thin film of GO is evenly covered on the surface of DE, I do not understand from what this statement can be deduced, seeming that images 1(a) and 1(b) are the same.
--- Images 1(a) and 1(b) are not the same. Form Fig. 1(b) the unevenness of GO can be clearly seen and GO covered on the surface of DE, with the holes preserved, which are even larger than before.
Reviewer 2 Report
The paper entitled “Study on morphology regulation of a diatomite-based catalyst and its catalytic mechanism on degradation of Ciprofloxacin” (Manuscript ID nanomaterials-2033423) by Ting Zhang , Jingjing Zhang, Yinghao Yu, Jinxu Li, Zhifang Zhou, Chunlei Li is in principle devoted to catalytic degradation of ciprofloxacin on selected metal oxides supported on composite carriers based on graphene oxide and diatomaceous earth. The proposed topic devoted to environmentally friendly degradation of ciprofloxacin belongs to the vital environmental issues particularly important for water purification. Even if the results presented in this paper may potentially be interesting for the scientific community, their presentation, interpretation and discussion are not acceptable in the current version of the paper. The Authors tried to couple too many issues in one paper, making it rather shallow, chaotic and eventually confusing for a reader. The article may be reconsidered for publication in Nanomaterials after improving by the Authors the points listed below.
Major remarks:
1. It is totally unclear what (in Authors' opinion) the most important catalytic component of the described MOx/GO-DE system, active in ciprofloxacin degradation, really is. In principle, it should be a metal oxide. It is thus not reasonable to put the emphasis in the title and in the text on diatomaceous earth, being just a part of the composite GO-DE support. The detailed speciation of the supported metal oxide entities should be discussed in detail as a function of metal nature and concentration as well as the effect of support composition on such speciation. The current state of the art related to the used metal oxides, which are active in ciprofloxacin degradation, is completely missing in the Introduction. Moreover, it is not clear why copper-based samples have been selected for further studies, when silver-based ones showed comparable activity in CIP decomposition as it can be seen in Fig. 1b.
2. Authors’ aim of work is not formulated clearly in this paper. In consequence the content and conclusions are too heterogeneous, and the issues are discussed in the chaotic way without focusing on one of the following problems: i) catalyst structure and its optimization towards the highest activity in ciprofloxacin degradation; ii) determination of active sites and morphology related effects and/or iii) crucial parameters determining degradation mechanism process. Conclusions are thus unspecific and should be rewritten regarding the aim of work.
3. There is no optimization of the catalytic system described in this paper. Contrary to this, the elements of optimization of CIP degradation conditions described in section 3.3 are beyond the scope of the paper.
4. Some terms used by the Authors in the text are too general, badly defined or unclear, e.g. “The main oxidizing substances in the catalytic system (…)” (p. 1, Abstract, line 11); “morphology regulation” (p. 1, Keywords, line 15), “(…) with high concentration” (p. 1, section1, line 21); “Heterogeneous catalysis (one of the advanced oxidation processes)” (p. 2, section 1, line 26); “radicals with strong oxidation” (p. 2, section 1, line 28); “high active silica” (p. 2, section 1, line 30); (…) sample surface being treated with Pt.” (p. 5, section 2.3, line 95); (…) the water sample (p. 7, section 2.6, line 135); (…) good adsorption (p. 7, section 3.1, line 140); “active substances (…) coated on GO-DE” (p. 8, section 3.1, lines 154-155); “load capacity” (p. 8, section 3.1, line 160); “(…) the characteristic spectrum of each element (…)” (p. 13, section 3.2, line 219);.
5. English usage should be checked by a native speaker to avoid incorrect, unclear sentences, like e.g. “(…) was magnetic stirred (…)” (p. 4, section 2.2, line 82); “Diatomite (DE) doesn’t have high specific area and good adsorption (…)” (p. 7, section 3.1, line 140); (…) the folded thin-film GO is evenly covered on the surface of DE (…)” (p. 10, section 3.2, lines 180-181); (…) which gives full play to the catalytic performance of the catalyst.” (p. 11, section 3.2, line 187); “(…) the Raman spectrum of GO mainly has the following two characteristics: (…)” (p. 11, section 3.2, lines 196-197); “(…) catalysts companying with H2O2 (…)” (p. 14, section 3.3, line 244 ).
6. The fragments devoted to degradation of MB should be removed from the text as not entirely relevant. They are confusing not only for a reader but even for the Authors, e.g. describing Fig. 1b labelled as “Influence of metal type loaded on GO-DE on CIP removal ration, the Authors’ comment is “the GO-DE support loaded with Cu and Ag have the best degradation ability of MB”.
7. Some references are missing in the text, e.g. the formula mentioned in the section 2.4 should also be referenced; “Graphene oxide (GO), first discovered in 1859, (…)” (p. 3, section 1, line 46); “For economic (…) catalyst preparation.” (p. 8., section 3.1., lines 159-160; references to XPS individual results (p. 13, section 3.2, lines 219-224);
8. It is not known what exactly the Authors concluded on catalyst structure basing on the results obtained by each of the individual techniques applied to characterize the investigated samples.
9. Interpretation of SEM images is far from a completeness: e.g. statistical elaboration and the related histograms are totally missing. In turn, interpretation of RS and FTIR results is unspecific and detached from speciation of the described surface entities. The statement “Raman spectroscopy is a typical technique to characterize the structure of GO” is simply not true.
10. Some statements are problematic or fully contradictory, e.g. “(…) GO is often used as a substitute for graphene.“ (p. 3, section 1, line 48); “It can be seen that preparation of CuO/GO-DE did not change the physical and chemical properties of DE.” (p. 12, section 3.2, lines 204-205): if it is so, why the described changes in the reactivity towards CIP are observed by the Authors; “(…) those peaks are characteristic diffraction peak of CuO. It is confirmed that copper has been successfully loaded in GO-DE carrier.” (p. 12, section 3.2, lines 207-208): CuO is not the same chemical substance as copper; the presence of CuO conformed by XRD cannot be interpreted in any other ways as the presence of nanocrystalline CuO over catalyst surface; on the other hand there are no peaks in the XRD patterns; the term “peak” is used for spectra and not for diffraction patterns where Bragg maxima can be observed.
11. Mechanistic considerations are unclear and rather tentative. Many Authors’ statements presented in this section are not supported by the reported experimental data. E.g. the presence of Cu(I) was not detected by any of the used characterization techniques. Moreover, the Authors mentioned active sites in their considerations without any reasonable definition what exactly they called by active sites.
Minor remarks:
1. used in the text abbreviation DE originates from diatomaceous earth and is not appropriate for the term "diatomite" preferred by the Authors;
2. Abstract is not informative enough for this paper content, among others just one active metal is mentioned; specific surface determination method and TEM should be mentioned in the Abstract too;
3. composition of the described samples should be clearly specified (each time both metal and graphene oxides contents should be indicated) in the text in sections 3.2- ;
4. an excessive comma should be removed from line 142;
5. the term “metal” cannot be used for metal ion;
6. hybridization type is usually denoted as sp2 and sp3 (as they come from the symbols of atomic orbitals) and not as SP2 and SP3 (p. 11, section 3.2, line 198);
7. results presented together in Fig. 3a-h should be separated reflecting individual techniques;
8. standard stability tests cannot be called continuous reusability experiments;
9. Fig. 6 is a type of graphical abstract and should not be placed directly in the text of the paper ;
10. it is not clear what the Authors mean by “calculation results” (p. 15, section 3.3, line 256);
11. the concentration of NaOH solution mentioned in the Experimental should be provided; potassium permanganate was not mentioned in section 2.1 Materials;
Author Response
18-Nov-2022
Manuscript ID: nanomaterials-2033423
Title: Study on morphology regulation of a diatomite-based catalyst and its catalytic mechanism on degradation of Ciprofloxacin
Nanomaterials
Responses to reviewer 2’s comments:
Thanks for your useful comments and suggestions on the language and the structure of our manuscript. We have modified the manuscript accordingly, and the detailed corrections are listed point by point in attachment.
